# Uptake of monoaromatic hydrocarbons during biodegradation by FadL channel-mediated lateral diffusion

Kamolrat Somboon[1,3], Anne Doble[2,3], David Bulmer[2], Arnaud Baslé[2], Syma Khalid[1✉] & Bert van den Berg [2✉]

In modern societies, biodegradation of hydrophobic pollutants generated by industry is important for environmental and human health. In Gram-negative bacteria, biodegradation depends on facilitated diffusion of the pollutant substrates into the cell, mediated by specialised outer membrane (OM) channels. Here we show, via a combined experimental and computational approach, that the uptake of monoaromatic hydrocarbons such as toluene in *Pseudomonas putida* F1 (*Pp*F1) occurs via lateral diffusion through FadL channels. Contrary to classical diffusion channels via which polar substrates move directly into the periplasmic space, *Pp*F1 TodX and CymD direct their hydrophobic substrates into the OM via a lateral opening in the channel wall, bypassing the polar barrier formed by the lipopolysaccharide leaflet on the cell surface. Our study suggests that lateral diffusion of hydrophobic molecules is the *modus operandi* of all FadL channels, with potential implications for diverse areas such as biodegradation, quorum sensing and gut biology.

[1] School of Chemistry, University of Southampton, Southampton SO17 1BJ, UK. [2] Biosciences Institute, The Medical School, Newcastle University, Newcastle upon Tyne NE2 4HH, UK. [3] These authors contributed equally: Kamolrat Somboon, Anne Doble. ✉email: s.khalid@soton.ac.uk; bert.van-den-berg@ncl.ac.uk

The production of pollutants in modern, industrialised society is a burden on the environment and poses substantial risks for human health because many of these compounds are toxic, carcinogenic and teratogenic[1–4]. One prominent class of pollutants are monoaromatic hydrocarbons (MAHs) such as BTEX (benzene, toluene, ethylbenzene and xylene), which naturally occur in crude oil. They are among the most abundant chemicals produced worldwide by the petrochemical industry, with benzene alone accounting for ~50 million metric tons. Due to their limited chemical reactivity and hydrophobic character, MAH and many xenobiotics (compounds foreign to life) are highly persistent within the environment. An additional concern with MAH is their relatively high aqueous solubility (~5 mM for toluene) and the risk that these compounds end up in drinking water[4]. Besides the obvious health-related factors, there is also a considerable economic burden from the presence of pollutants within the environment, due to the high costs of the treatment and cleanup of contaminated sites by various abiotic methods. These factors have generated an enormous interest in the metabolisation of pollutants by bacteria (biodegradation), and the utilisation of such bacteria in the removal of pollutants (bioremediation)[5–7].

MAHs are predominantly degraded by aerobic soil bacteria, which utilise these compounds as sole sources of carbon and energy. Many MAH biodegraders are Gram-negative bacteria, and genera with an abundance of biodegrading bacteria include *Sphingomonas* (α-proteobacteria), *Burkholderia* (β-proteobacteria), and *Pseudomonas* (γ-proteobacteria). *Pseudomonas putida* F1 (*Pp*F1) was isolated from a polluted creek in Urbana (Illinois, USA), and was the first biodegrader whose genome was sequenced (https://genome.jgi.doe.gov/portal/psepu/psepu.info.html). *Pp*F1 has a large genome (6.0 Mb) and is a versatile organism found in soil and water that can grow on benzene, toluene, ethylbenzene and p-cymene (isopropyltoluene) as sole sources of carbon (Supplementary Fig. 1)[8–11]. A practical advantage of *Pp*F1 is that its biodegrading capabilities are encoded on the genome and not on large catabolic plasmids. *Pp*F1 has two operons devoted to the biodegradation of MAH, *tod* and *cym/cmt*, adjacent to each other on the genome (Supplementary Fig. 1)[12].

The first step in MAH metabolism is the introduction of oxygen into the aromatic ring, carried out by a multi-component dioxygenase that has been widely used in biocatalytic syntheses of chiral chemicals[13,14]. This step, as well as the subsequent enzymatic steps in MAH degradation, occurs in the cytoplasm. Since the plasma membrane is permeable for hydrophobic molecules (hydrophobics)[15,16], biodegradation in Gram-negative bacteria depends on the presence of diffusion channels in the outer membrane (OM), which does form a barrier for hydrophobics due to the lipopolysaccharide (LPS) in the outer leaflet[17].

Thus far, only the FadL OM channel family has been shown to mediate transport of hydrophobics[18,19]. FadL from *E. coli* (*Ec*FadL) is the archetype of the family and is a long-chain fatty acid (LCFA) transport channel[20–22]. The lumen of the 14-stranded β-barrel of *Ec*FadL is occluded by an N-terminal ~40-residue plug domain[23], reminiscent of the plug or cork domain of much larger TonB dependent active transporters. A lateral diffusion transport mechanism for LCFA uptake by *Ec*FadL has been established via detailed structure-function studies[22]. The key structural feature in *Ec*FadL is an opening in the side of the barrel wall caused by an inward kink in β-strand S3. LCFA binding in an adjacent high-affinity site causes a conformational change in the plug N-terminus, opening up a diffusion pathway through the lateral opening into the OM[22–24].

With respect to MAH uptake, the crystal structure of the putative toluene channel TodX from *Pp*F1 has been determined previously and is similar to that of *Ec*FadL in aspects such as the presence of a plug domain and a lateral opening at the same position as in *Ec*FadL[25]. However, TodX also has a classical, polar channel through the plug domain that leads directly into the periplasmic space. This feature, coupled with the much higher aqueous solubilities of MAH compared to LCFA (mM vs. nM), raises the possibility that MAH transport occurs via a classical mechanism like that used by most other OM channels, i.e. directly into the periplasmic space[25].

Here we show, by using an integrated approach combining growth assays, X-ray crystallography and molecular dynamics (MD) simulations, that MAH uptake as the first step in biodegradation does occur via lateral diffusion. The fact that uptake of hydrophobics with very different structural and physicochemical properties (i.e. LCFA and MAH) occurs via the same mechanism suggests that lateral diffusion is the *modus operandi* for all FadL-family channels and their large ensemble of potential transport substrates, with potential implications for large areas of Biology. In addition, FadL channels could be utilised to develop genetically engineered biodegrading bacteria that are more effective in the uptake of pollutants or have wider substrate specificities. Applications of FadL channels could also be envisioned for biosensor and biocatalyst bacterial strains for more efficient detection of pollutants and for better syntheses of pharmaceuticals and other industrial products[26].

## Results

**TodX and CymD are MAH uptake channels.** *Pp*F1 has three FadL orthologs, of which TodX (*Pput_2883*) and CymD (*Pput_2901*) are located in the adjacent *tod* and *cym/cmt* operons respectively. The third ortholog (*Pput_4030*) is not in an operon and, since it is most similar to *Ec*FadL (29% sequence identity), will be referred to as F1FadL. To enable structure-function studies we constructed the triple knockout strain *Pp*F1-3, with clean deletions for *todX*, *cymD* and *F1fadL* (Δ*F1fadL*Δ*todX*Δ*cymD*). To link FadL channels to MAH uptake, a variety of assays in liquid culture were tested, including bioluminescence in the *Pp*F1 reporter strain *Pp*G4[27] and its triple knockout *Pp*G4-3. All these assays were very challenging, being hampered either by MAH-related toxicity issues or an inability to observe reproducible differences between the wild-type and triple knockout strains. In addition, uptake assays using radiolabeled substrates as used for LCFA uptake by *Ec*FadL[22,24] are clearly impractical for volatile compounds such as MAH. We therefore opted to grow various strains on minimal medium agar plates with toluene or p-cymene vapour as the sole carbon source (Methods), and report colony size as a semi-quantitative measure for growth. As a control, growth was monitored with glucose as the sole carbon source, showing that wild-type *Pp*F1 and the triple knockout *Pp*F1-3 grow equally well (Fig. 1a). In the case of MAH, both for toluene and p-cymene there are clear differences in growth between wild type and the triple knockout (Fig. 1b–e). *Pp*F1-3 still grows slowly on MAH vapour, most likely as a result of spontaneous (i.e. non protein-mediated) OM diffusion. A further deletion of *oprG* and *tcpY*, other potential hydrophobics uptake channels[28,29], in the *Pp*F1-3 strain to generate *Pp*F1-5, did not reduce the background (Fig. 1). Toluene causes the most prominent background growth (Fig. 1a, b), most likely due to its smaller size and higher volatility compared to p-cymene. The background colonies with p-cymene have similar diameters compared to those with toluene but are more translucent, possibly because they are thinner, generating very clear phenotypes on imaged plates (compare Fig. 1b, c).

To complement *Pp*F1-3, we used the bacterial Tn7 transposon to integrate various alleles at the single *att*Tn7 site of the bacterial chromosome, under the control of the $P_{BAD}$ promoter (Methods).

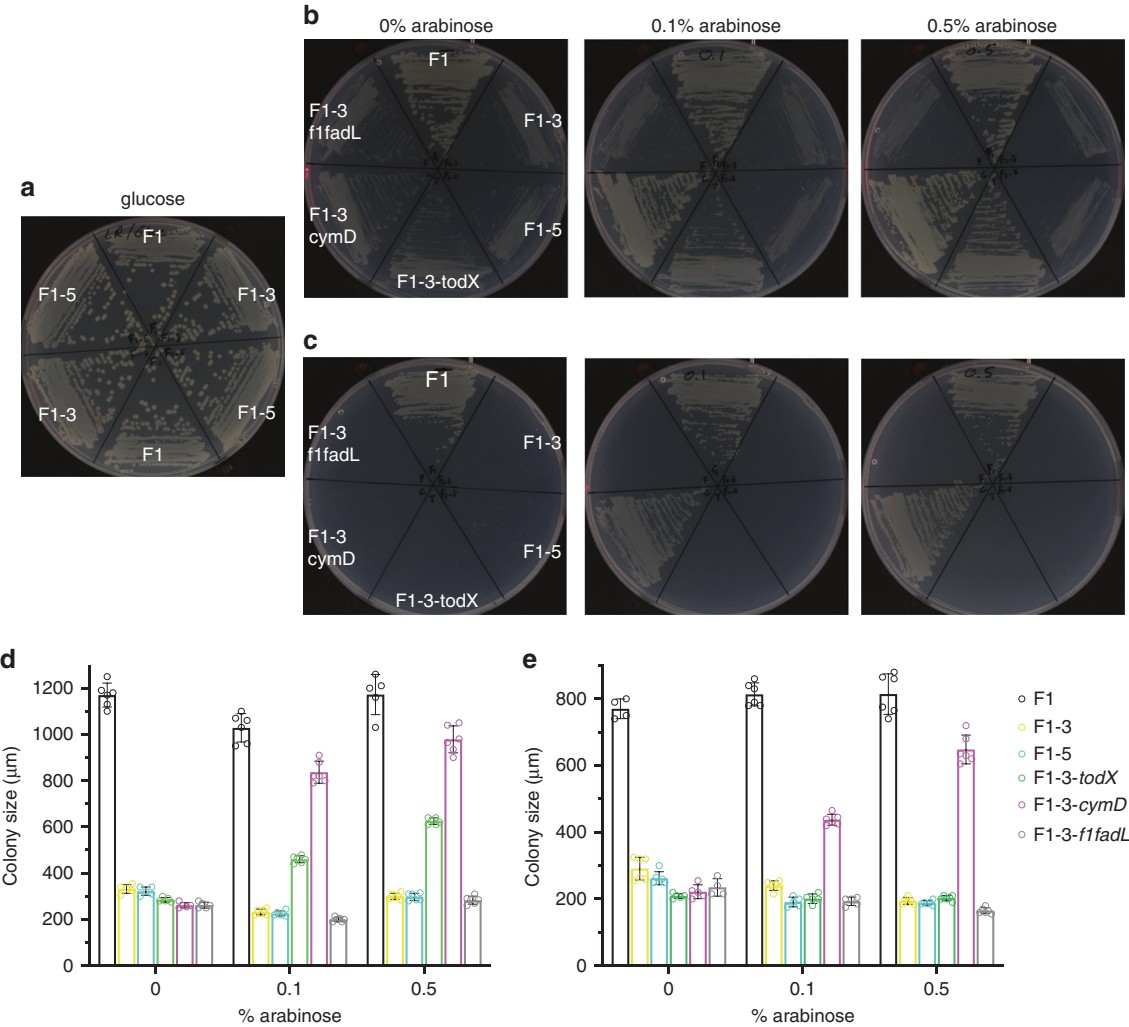

**Fig. 1 TodX and CymD are MAH uptake channels. a** Growth of *Pp*F1, *Pp*F1-3 and *Pp*F1-5 on 0.5% glucose. **b**, **c** Growth of various *Pp*F1 strains with and without Tn7 complementation on minimal medium with vapour-phase toluene (**b**) or p-cymene (**c**) as the sole carbon source. Growths shown are representative of 2 or 3 plates. **d**, **e** Individual maximum colony sizes from two independent plates after (**d**) 4 days growth on toluene at 20 °C or (**e**) 5 days on p-cymene at 30 °C with different arabinose concentrations. Reported values are mean ± s.d ($n = 5$–8). Bar graphs and dot plots were generated in GraphPad Prism version 7. Source data for colony sizes are provided in the Source Data file.

In the absence of arabinose, only the wild-type *Pp*F1 strain grows, and none of the complemented strains grow better than the *Pp*F1-3 background (Fig. 1b, c). Both TodX and CymD complement growth of *Pp*F1-3 on toluene in the presence of 0.1% arabinose (Fig. 1b). Increasing the arabinose concentration to 0.5% gives slightly better growth relative to 0.1%, but only for *Pp*F1-3-*todX* and *Pp*F1-3-*cymD*, demonstrating that even relatively high concentrations of the sugar by itself do not allow growth. There is no complementation with *Pp*F1-3-*F1fadL*, indicating that LCFA channels do not allow MAH uptake and strengthening previous data, based on lack of LCFA uptake by TodX, that FadL channels are substrate specific[25]. Interestingly, TodX and CymD differ in their ability to complement growth on MAH. Compared to *Pp*F1-3-*cymD*, *Pp*F1-3-*todX* grows slightly slower on toluene and clearly does not grow on p-cymene (Fig. 1b–e). This suggests that even closely related MAH channels have different specificities, with CymD being able to transport more bulky substrates, such as p-cymene.

**Molecular dynamics simulation reveal an MAH binding site.**
Extensive attempts to identify an MAH binding site within TodX via experimental approaches failed, including the use of non-volatile heavy atom MAH derivatives (e.g. iodotoluene) in co-crystallisation and soaking. Mixed micelles in the presence of mM concentrations of MAH derivatives have very different properties than those of pure detergent, leading to phase separation problems. In addition, the $C_8E_4$ crystallisation detergent is present at relatively high concentrations (~13 mM) and will bind preferentially to the hydrophobic areas inside TodX. To obtain information about substrate binding, we therefore performed equilibrium MD simulations with TodX embedded in a physiological, asymmetrical OM, with benzene used as the substrate (Methods). Besides being a physiological substrate for *Pp*F1[9], benzene has the additional advantage for MD in that it is small and symmetric. We placed three benzene molecules near the extracellular end of the hydrophobic channel that was previously identified in TodX based on bound detergent[25] (Extended Data Fig. 2). Two independent simulations were performed and showed that benzene can enter the protein readily (1 and 2 benzenes entered in the two independent simulations respectively). In one simulation, the lone benzene entering the channel became restricted to a fairly well-defined region close to the N-terminus after ~50 ns. During the simulation in which two

**Fig. 2 An N-terminal conformational change generates a high-affinity binding site. a** Movement of a single benzene within the unrestrained wild-type protein, corresponding to the simulation shown in Supplementary Fig. 2a. The benzene becomes confined to the P-pocket after the protein undergoes a conformational rearrangement to state II, after about 60 ns. The surface of the plug domain (yellow) is shown to emphasise the difference in conformation. **b** Benzene is mobile with the N-terminus of the protein restrained in state I (left panel), whereas restraining the N-terminus in state II rapidly confines the benzene to the P-pocket. **c** Benzene freely diffuses through the protein in simulations of the N-terminal deletion mutant. All simulations show benzene molecules from 250 equally spaced frames, according to the colour scheme indicated (0 ns, red; 250 or 500 ns, violet).

benzenes entered the protein, the first benzene to enter remained near a similar region of the protein, but not as close to the N-terminal domain and was not restricted to the same extent as the lone benzene in the first simulation. The second benzene initially diffused freely, but then also rapidly moved into the region occupied by the first benzene, such that after 100 ns the two benzene molecules simultaneously occupied a region near the N-terminus and remained there (Supplementary Fig. 2).

We next performed a cluster analysis on the trajectories to identify dominant conformational states of the protein during the simulations. This analysis enabled the identification of two distinct conformational states of the N-terminus (Supplementary Fig. 3). The first state (state I) corresponds to that of the crystal structure, with the α-amino group of Thr1 close to Arg136. Following a conformational change, the Thr1 α-amino group moves more than 10 Å such that it comes close to Leu105 (Supplementary Fig. 4). This conformation is stabilised through interactions between Ser103, Leu105, Pro371 and Thr1 and between Pro371 and Gln2. Interestingly, this state (state II) appears to confine the benzene molecule to a fairly well-defined binding pocket close to the N-terminus, which we designate as the principal binding pocket (P-pocket). To further investigate the effect of different conformational states of the N-terminus on benzene dynamics, we performed additional MD simulations, but with the N-terminus restrained to either state I or state II. With the N-terminus occupying state I, the benzene molecule is extremely mobile and diffuses back and forth (Fig. 2), without preference for any particular part of the protein. By contrast, with the N-terminus in state II, the benzene molecule diffuses rapidly towards the N-terminus and remains in the P-pocket. Thus, these simulations suggest that a specific conformation of the N-terminus is required for high-affinity substrate binding.

This hypothesis is supported by a simulation where the N-terminal 4 residues are removed, and in which the benzene remains highly mobile (Fig. 2c). This observation is reminiscent of *Ec*FadL, where deletion of the first 3 residues of the N-terminus causes a disruption of the high-affinity LCFA binding site and a loss of transport activity[24]. Analogous to Phe3 in *Ec*FadL, the TodX side chain of Gln2 in state II is part of the P-pocket, which is additionally defined by the side chains of Leu132, Leu162, Leu164, Leu166, Val272, Phe275, Val313, Leu326 and Ile370 (Fig. 3).

**Lateral opening mutants are defective in MAH uptake.** Previously, we demonstrated the lateral diffusion mechanism for LCFA

transport by *Ec*FadL via site-directed removal of the lateral opening, by flattening the inward-kinked part of strand S3 (Fig. 4)[22]. Crucially, closing off the *Ec*FadL lateral opening did not lead to any other changes in the protein structure, and expression levels of the mutants were similar as for wild type. We therefore opted to use the same strategy for TodX and CymD, and reasoned that a lack of growth on MAH after closing the lateral opening would provide strong evidence for a lateral diffusion mechanism. The alternative experiment, i.e. closing the classical channel through the TodX plug domain[25], would likely be much more difficult given that the plug of *Ec*FadL does not tolerate changes well.

The lateral opening in TodX involves both β-strand S2 and S3, contrasting with *Ec*FadL in which only strand S3 is involved. Strand S2 in TodX bulges outward, whereas S3 has a less pronounced kink in the same place as *Ec*FadL (Fig. 4). To flatten the S2 bulge, we removed seven residues that were not visible in the density of TodX, replaced a proline with alanine and inserted an alanine to connect the strands (Methods; ΔS2). To flatten the S3 kink in the ΔS2 mutant, we removed the conserved glycine residue G104 to give ΔS2S3. We determined the X-ray crystal structures of both TodX variants which showed that, as expected, the size of the lateral opening is drastically reduced as a result of the flattening of the strands (Fig. 4, Supplementary Fig. 5 and Supplementary Table 1), while the remainder of the mutant structures are very similar to those of wild type.

To enable similar experiments on CymD, we first determined its crystal structure. Despite considerable efforts, we could only obtain one crystal form that diffracted to sufficient resolution for structure determination (Supplementary Table 1). The CymD structure is very similar to that of TodX but lacks density for ~30% of the protein due to mobility of several extracellular loops within the crystal (Supplementary Fig. 5). Importantly however, the visible parts of the structure show that the S2 bulge and the S3 kink are present and in the same location as in TodX. To allow molecular dynamics simulations, we modelled the missing parts of CymD by using the TodX structure as a template (Fig. 4; Methods). For site-directed mutants, we generated ΔS2 in the same way as for TodX. To assess the importance of the S3 kink, the ΔS3 variant was made by deletion of the conserved residue G105. The final CymD mutant was G105A, made to assess the importance of the invariant glycine at this position. Due to the difficulties in crystallising wild-type CymD and its similarity to TodX, no attempts were made to crystallise CymD lateral opening mutants.

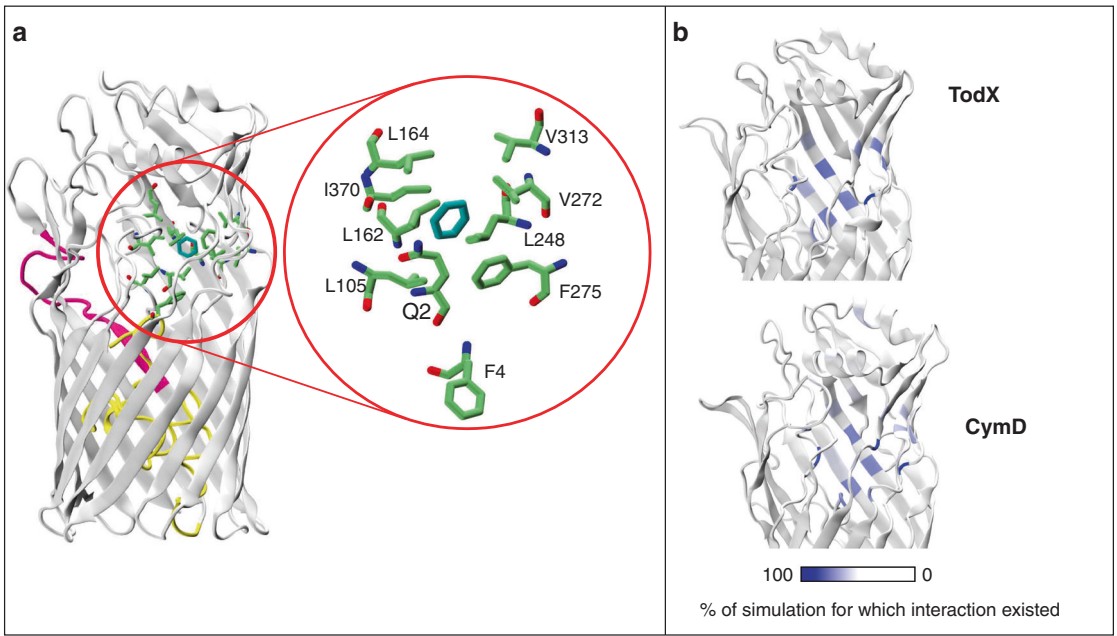

**Fig. 3 MD simulations reveal binding pockets for MAH in TodX and CymD. a** Cartoon model showing state II TodX residues mediating the largest number of interactions with benzene (blue) from equilibrium MD simulations as green stick models. These residues form the P-pocket. **b** Comparison of the location of residues in TodX and CymD that form the largest number of interactions with benzene and p-cymene respectively. Residues are coloured by the percentage of simulation time for which the interactions occur, with data included for both independent simulations of each protein. Interactions are defined as an inter-atomic distance of <4.0 Å.

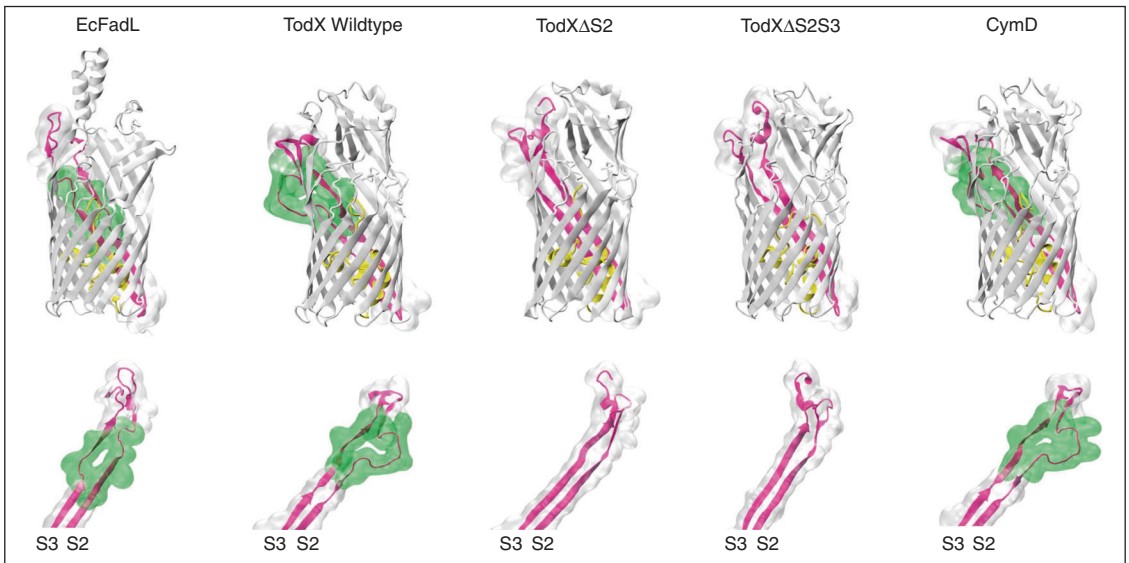

**Fig. 4 Lateral opening mutants of TodX.** Structures of the FadL channels discussed in this study. The lateral openings (delineated by the green surface) are similar in size in the wild-type structures of TodX and CymD and differ only slightly from EcFadL, in that the latter does not have a bulge in S2 like TodX and CymD, but a more pronounced kink in S3. The two TodX mutant channels ΔS2 and ΔS2S3 have much smaller lateral openings, while retaining overall high structural similarity to the wild-type protein.

Plate growth assays with vapour-phase toluene showed background growth for the TodX and CymD lateral opening mutants, indicating that they are defective in toluene uptake (Fig. 5a, b). For TodX, background growth is already observed for the ΔS2 mutant, so the effect of the additional flattening of strand S3 in the ΔS2S3 variant cannot be assessed. For CymD, the ΔS2 and ΔS3 mutants both cause background growth, indicating that the relatively minor change of flattening S3 is sufficient to impede lateral diffusion (Fig. 5a, b). As mentioned above, growth on p-cymene vapour is observed only with CymD, and the CymD lateral opening mutants have the same phenotypes with toluene and p-cymene. Interestingly, the invariant glycine residue in the S3 kink is not essential for function, as growth for the G105A variant is only slightly lower than that mediated by wild-type CymD (Fig. 5b).

Since growth will depend on the amounts of the various MAH channels in the OM, it was important to determine relative expression levels. For this, we used the broad-host-range pHERD30 plasmid to complement PpF1-3 with the complete set of C-terminally His-tagged *cymD* alleles via induction with

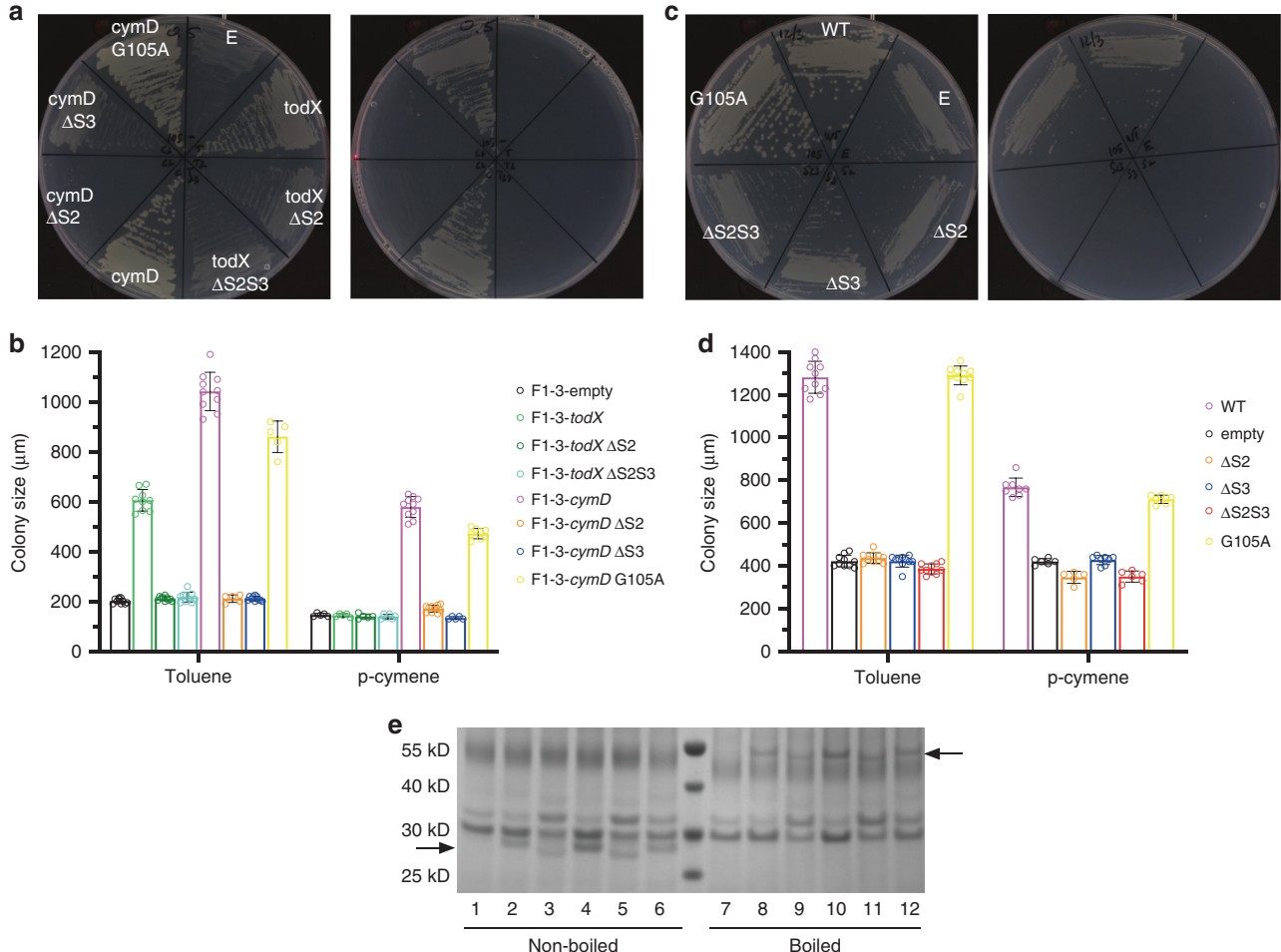

**Fig. 5 Lateral opening mutants of TodX and CymD have impaired MAH uptake. a** Representative growth of Tn7-complemented *Pp*F1-3 strains on toluene (left panel) and p-cymene vapour in the presence of 0.5% arabinose. Growths shown are representative of 2 or 3 plates. **b** Corresponding maximum colony sizes from (**a**), taken from two different plates (mean ± S.D; $n = 6$–10). **c** Representative growth of pHERD30-complemented *Pp*F1-3 strains on toluene (left panel) and p-cymene vapour in the presence of 0.1% arabinose. Growths shown are representative of 2 or 3 plates. **d** Corresponding maximum colony sizes from **c**, taken from 2 different plates (mean ± S.D; $n = 6$–10). **e** Coomassie-stained SDS-PAGE gel of pHERD30-complemented *Pp*F1-3 strains after IMAC. lanes 1/7, empty plasmid; lanes 2/8, CymD wild type; lanes 3/9, CymD ΔS2; lanes 4/10, CymD ΔS3; lanes 5/11, CymD ΔS2S3; lanes 6/12 CymD G105A. Arrows indicate approximate positions of CymD variants before (lanes 1–6) and after boiling (lanes 7–12). As is the case for many OMPs, CymD remains partially folded in SDS without boiling, leading to faster migration in the gel and a lower apparent molecular weight (~31 kD compared to ~48 kD after boiling). Molecular weight markers are indicated on the left. The gel picture shown is representative of two independent expression analyses. Source data for colony sizes are provided in the Source Data file.

arabinose. We focused on CymD since its OM levels are higher than TodX, making bands on SDS-PAGE easier to see. For the plasmid-based expression, 0.5% arabinose results in poor growth for all variants, most likely due to toxicity effects associated with relatively high levels of OMP overexpression, and therefore 0.1% arabinose was used. The observed phenotypes are very similar for chromosomal and plasmid-based complementation (Fig. 5a–d). Importantly, the OM protein levels for all CymD mutants are similar to those of the wild type (Fig. 5e), indicating that the loss-of-growth phenotypes are not due to lack of protein in the OM, and supporting the hypothesis that MAH are taken up by *Pp*F1 via lateral diffusion.

**MD simulations favour lateral diffusion over classical transport.** To provide additional support for lateral diffusion we next carried out a series of umbrella sampling and steered MD simulations. In order to provide an energetic explanation of where the barriers to permeation through the classical route are

located, we first calculated the free energy profile for benzene permeation through TodX via the classical route using umbrella sampling. The free energy profiles are normally constructed by setting the bulk water regions on the extracellular and periplasmic sides equal to 0 kT. However, inspection of the non-symmetrised free energy profile shows a large difference of ~25 kT between the free energies of the extracellular and periplasmic sides (Supplementary Fig. 6). Extension of the trajectories of umbrella sampling windows by a further 50 ns did not lead to the closing of this energy gap. Therefore, we have not used the umbrella sampling for drawing conclusions about the energetics of our system.

We next employed steered MD to 'pull' benzene along the classical route. The resulting force profile identified the location of barriers for benzene permeation. The profile shows two barriers encountered en route to the periplasmic mouth of the protein, at $z \sim -3$ and $z \sim 20$–25 Å (Fig. 6). The latter corresponds to the region containing the plug domain. Given these barriers would have to be overcome by a benzene molecule attempting to permeate through the protein, their location

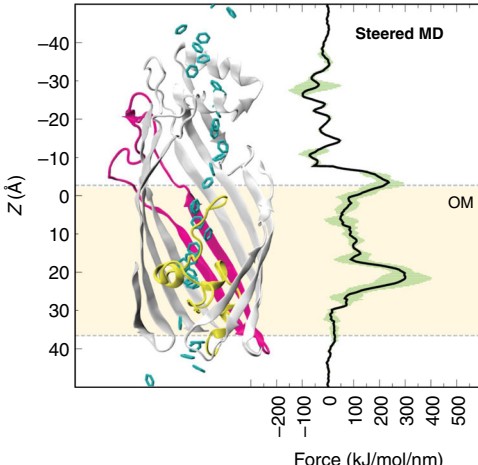

**Fig. 6 Steered MD reveals barriers for benzene permeation.** An overlaid force profile from the steered MD simulations via the classical pathway (right) on the cartoon representation of TodX, with benzene positions shown along the reaction coordinate (centre). The TodX plug domain is coloured yellow, S2 and S3 are coloured magenta. Standard deviations are shown in green.

indicates that the classical pathway may not be favourable for benzene translocation into the periplasmic space. Next, steered MD simulations to pull benzene through TodX with the rearranged N-terminus (state II) were performed. Interestingly, the resultant force profile reveals the first energetic barrier to be absent and consequently, regions that previously were energetically separated are now more connected (Supplementary Fig. 7). However, the second, larger energy barrier inside the plug domain is unchanged, indicating that classical diffusion remains unfavourable. We note that while the steered MD simulations are reliable in showing large energetic barriers, they are less reliable in reproducing the fine details of energy wells. It is therefore difficult to evaluate, for example, whether the region at the mouth of the lateral gate ($Z \sim 0$) is lower in energy than the surrounding area using steered MD (Fig. 6).

As detailed above, the rearrangement of the N-terminus to state II confines the benzene to the P-pocket around $Z -10$ Å, and removes the first energetic barrier at $Z \sim -10-0$ Å in the force profile constructed from steered MD simulations. However, the force profile is still increasing, making it unlikely that diffusion to the lateral opening would happen spontaneously. Indeed, benzene is very stably confined to the P-pocket in our equilibrium simulations, and how it gets to the lateral opening at $Z \sim 0$ is presently unclear. At the lateral opening, benzene would have three options compared to permeation along the classical route; (i) remain inside the protein, (ii) leave the protein via the same route it entered or (iii) leave the protein via the lateral gate and thereby enter the OM. We used a combination of six steered MD and six equilibrium MD simulations to explore these possibilities. In all of these simulations, benzene was initially located at the mouth of the lateral opening, at a distance of ~9 Å from its position in the P-pocket (Fig. 7). This location of benzene was taken from the original benzene position in one of the umbrella sampling simulations in which benzene was restrained at $z = 1.6$ Å, (as noted above, the free energy profile is presented in the Supplementary Fig. 6, but not discussed here due to lack of convergence). In this simulation, benzene was observed to spontaneously diffuse ~6 Å to the mouth of the lateral opening, and the equilibrium simulations were therefore initiated from this point. Overall, the steered MD and equilibrium simulations

revealed three pathways for benzene exit via the lateral opening (Fig. 7).

Pathway A was identified from steered MD simulations, pathway B was observed in steered MD and equilibrium MD simulations, and pathway C was observed only in equilibrium MD simulations. In pathway A, observed in three of the six steered MD simulations, benzene remains on the S3 side of the protein. In pathway B, observed in the other three steered MD simulations, the benzene moves through the S2 bulge and thus ends up on the S1 side of the bulge (Fig. 7a). As a result, the benzene completely avoids the polar regions of LPS and enters the hydrophobic region of the outer leaflet, facilitated by slight widening of the lateral opening (Fig. 7a). In contrast, by not going through the S2 bulge in pathway A, the benzene is forced towards the sugars of LPS. Inspection of the force profiles (Fig. 7b) shows that, as expected, pathway B is more favourable than pathway A. Regarding the six equilibrium MD simulations, in two of these benzene moved out of the protein via pathway C. In this pathway, benzene interacts with Tyr79 initially, and then moves towards S3. Subsequently, a widening of the lateral opening enables it to then move out into the OM. In the other four equilibrium simulations, benzene exited the protein via pathway B in two simulations. In one simulation, it remained very close to its initial location near the lateral opening. Interestingly, in the final simulation the benzene moved from the lateral gate back into the high-affinity binding site (P-pocket) in the protein interior. After the benzene diffused back into the P-pocket, the N-terminus of the protein switched to state II.

**Transport of p-cymene by CymD is similar to benzene transport by TodX.** So far, we focused on benzene in our simulations due to its small size and symmetry. However, given that our experimental data clearly support the diffusion of p-cymene through CymD, we next asked if the simulations could also capture the diffusion pathway of this more complex substrate. To explore this, we simulated CymD with p-cymene, in a similar manner to the equilibrium simulations of TodX with benzene described above. Two independent equilibrium MD simulations with the p-cymene starting position close to the channel entrance show that p-cymene behaves in a similar manner as benzene in TodX, in that upon entering the protein it quickly becomes confined to the P-pocket and remains there for the duration of the $2 \times 500$ ns simulations (Fig. 3 and Supplementary Fig. 8). Interestingly, while in the P-pocket, p-cymene showed no orientational preference (Supplementary Fig. 8). In contrast to TodX, we did not observe a "state II-like", large conformational rearrangement in the N-terminus in order for p-cymene to be confined to the P-pocket of CymD, perhaps due to its larger size.

To assess the importance of the P-pocket in CymD, we performed mutagenesis studies using the pHERD30 system described above. Given that the pocket is quite large and that the substrate occupies a substantial area (Fig. 3), it is difficult to gauge which residues to mutate and what kind of mutations to introduce. We opted to make large changes, and focused on the residue pairs L164D/L166D and L273N/F276D (methods), which correspond to L162/L164 and V272/F275 in TodX (Fig. 3a). Growth of the variants was assessed both on toluene and p-cymene, and expression levels were determined via SDS-PAGE. Both P-pocket variants are present in the OM at much lower levels than WT, suggesting that their biogenesis is compromised (Fig. 8), a finding which perhaps is not surprising given the large changes. Despite this, L273N/F276D showed robust growth on both toluene and p-cymene, suggesting (i) that factors other than OM transport are rate limiting for

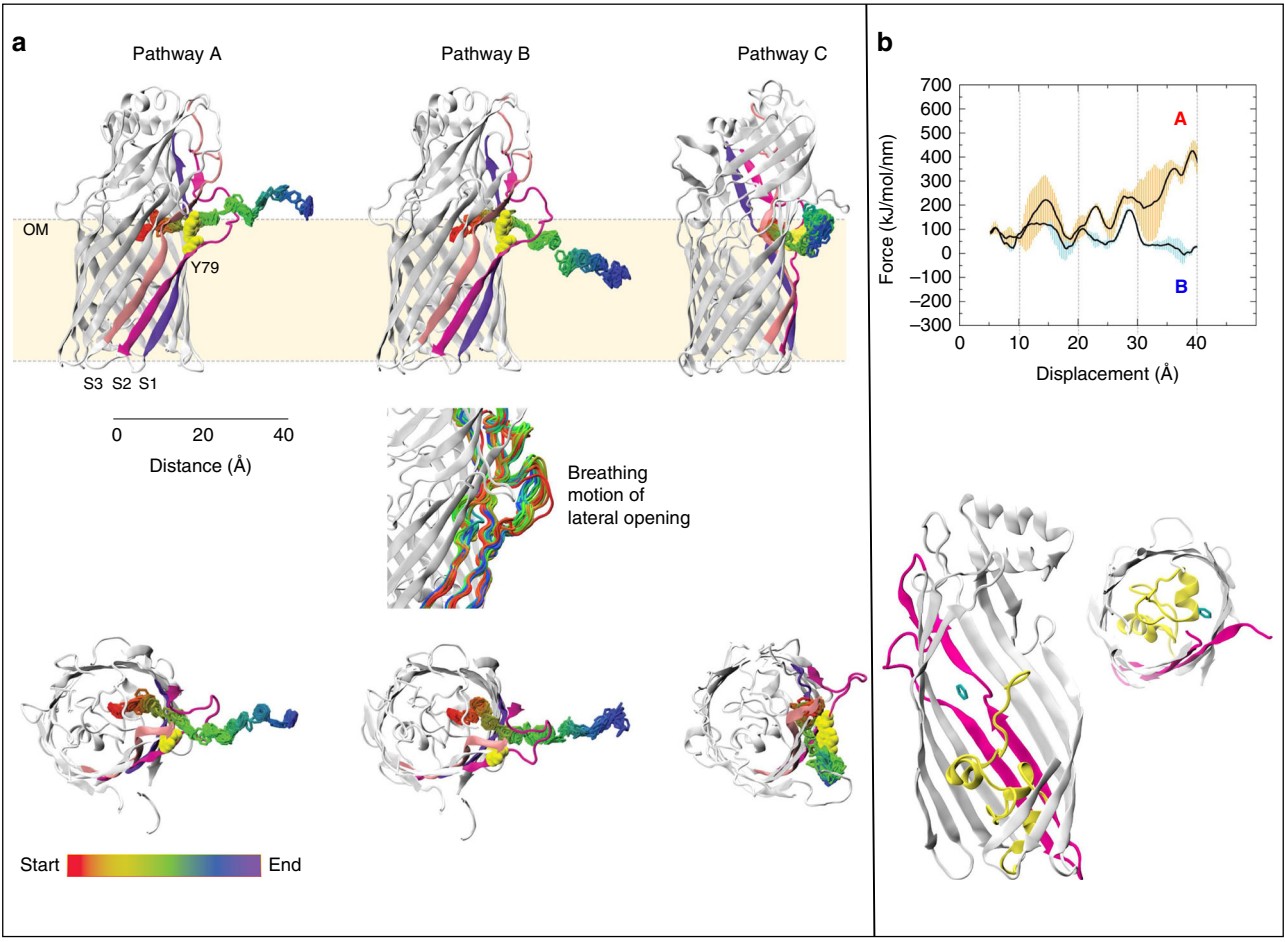

**Fig. 7 Identification of three distinct benzene pathways into the OM through the lateral opening in TodX. a** Views from the side (top panel) and from the extracellular side (bottom panel), showing locations of benzene molecules from 250 equally spaced frames for each simulation, according to the colour scheme indicated. The middle panel shows the breathing motion of the lateral opening during benzene exit via pathway B. **b** Force profiles corresponding to pathways A and B (top panel) and views of the initial location of benzene for the equilibrium simulations (bottom). Standard deviations are shown in orange and blue.

growth on MAH, and (ii) that the L273N/F276D mutant is most likely fully functional. Lower levels of growth above the background are observed for L164D/L166D (Fig. 8), suggesting this mutant is also functional. Thus, P-pocket mutations do not have strong effects on MAH diffusion.

Given that we do not observe spontaneous diffusion of benzene from the P-pocket to the lateral opening in TodX, we expected p-cymene to behave similarly within CymD. Therefore, in order to study potential diffusion of p-cymene out of the protein, six equilibrium simulations were initiated from p-cymene placed at the lateral opening of CymD, based on the position of benzene in TodX from umbrella sampling simulations. P-cymene diffused laterally out of the protein in three out of six simulations (Supplementary Fig. 9). In two of these simulations, p-cymene occupied a position just on the outside of the protein but remained close to the lateral opening after 250 ns. Upon extension of these two simulations, p-cymene moved into the OM after ~480 ns in one case, whereas in the other it remained outside, but interacting with the lateral opening of CymD even after 500 ns. These results indicate that lateral diffusion from CymD into the OM is readily feasible for a substrate larger than benzene. In all three simulations, p-cymene exit was via the same pathway. Finally, we performed steered MD simulations for p-cymene in CymD to characterise both the classical and lateral diffusion pathways. The force profile for the classical route closely

resembles that for TodX in state II, i.e. there is no large barrier for p-cymene movement past the lateral opening, but there is a sizeable barrier for moving through the plug domain (Supplementary Fig. 10). In other words, diffusion via the classical route is unfavourable. Force profiles from steered MD simulations in which p-cymene is pulled out of the lateral opening are also similar to those for benzene being pulled out of TodX, but with the force barriers around the lateral opening being somewhat higher and wider (Supplementary Fig. 10). The pathway of p-cymene exit from these simulations was the same as that observed from the three equilibrium simulations in which spontaneous exit occurred. Thus, we only observe one pathway for lateral exit of p-cymene from CymD. The pathway is similar to pathway A observed for benzene exiting TodX, in terms of its relationship to the S2 bulge.

A key difference between CymD and TodX is that TodX does not grow on p-cymene. We therefore ran steered MD simulations to explore the feasibility of p-cymene exiting from the lateral opening of TodX. As for benzene, two pathways for p-cymene exit were identified (Supplementary Fig. 11). The force profiles show that pathway B is preferred for both substrates, but the required force for p-cymene is approximately twice that required for exit of benzene. This result suggests that the inability of TodX to support PpF1 growth on p-cymene may be due to properties of the TodX lateral opening.

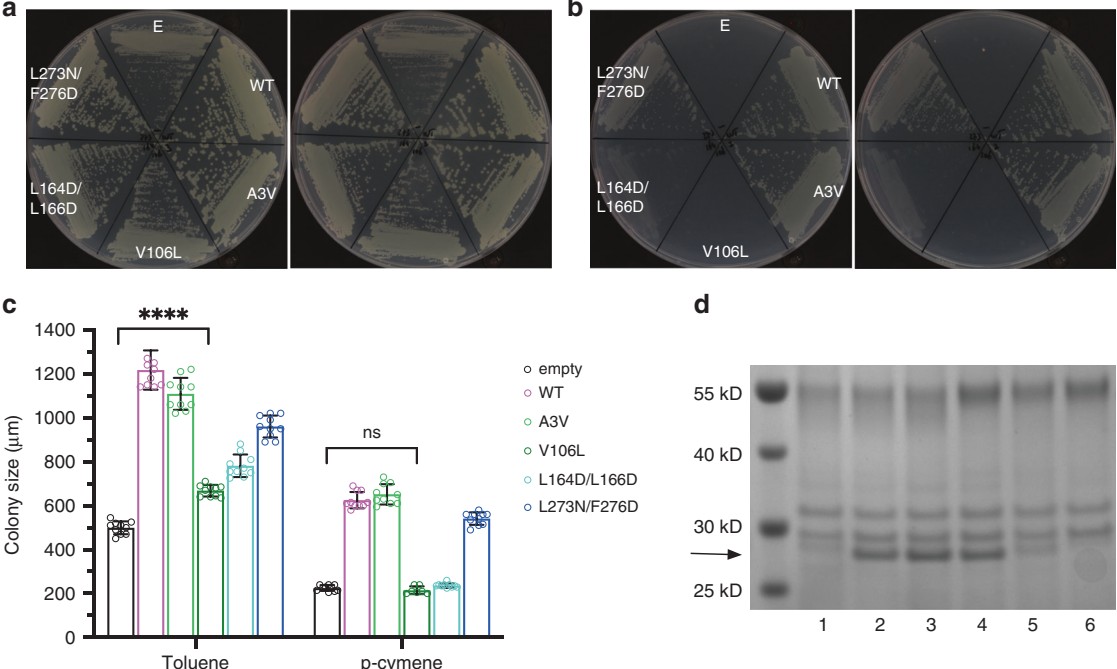

**Fig. 8 Structure-function studies of CymD. a, b** Growth on toluene (**a**) and p-cymene (**b**) for *Pp*F1-3 complemented with pHERD30-expressed *cymD* variants. Growths shown are representative of 2 or 3 plates. **c** Quantitation of growth via measurement of maximum colony sizes from (**a**) and (**b**) (mean ± S.D; $n = 6$–10). Significance levels were analysed via two-tailed Student's *t* test. n.s., not significant ($p = 0.2$); ****, $p < 0.000001$. **d** SDS-PAGE showing OM expression levels of the various strains. The location of CymD is shown by the arrow. Only boiled samples are shown. Source data for colony sizes are provided in the Source Data file.

## Discussion

The fact that toluene, with its small size and hydrophobic character, permeates poorly through the OM of *Pp*F1 in the absence of MAH channels emphasises the truly remarkable permeation barrier formed by this ~30 Å wide hydrophobic layer covered by cross-linked sugars on the extracellular surface. In the presence of TodX/CymD, robust growth is observed, providing the first direct evidence for MAH transport by FadL-type channels[30,31]. Our equilibrium MD simulations show very rapid movement of MAH molecules over a distance of ~30 Å, from the extracellular surface of the protein, via a largely hydrophobic tunnel, to a well-defined high-affinity binding site close to the N-terminus. This shows that MD is particularly well-suited to study the protein-mediated diffusion of small hydrophobic molecules, while experimental methods are challenging. Within the P-pocket, the MAH molecules appear to be quite stable, and it is currently not quite clear how the MAH moves from the P-pocket to the lateral opening. The limitation of equilibrium MD timescales is not allowing us to observe this movement, and likely requires enhanced sampling methods in follow-up studies. However, the TodX simulation where benzene moves from the lateral opening back to the P-pocket is informative. Here, the benzene moves ~9 Å in a very short time (~30 ps; Supplementary Fig. 12). The movement to the P-pocket requires passage through a narrow, hydrophobic "sphincter" formed by the side chains of Val3, Leu105 in the S3kink, and Ile370 in loop L6. Perhaps surprisingly, there are no dramatic conformational changes, with minor, "breathing" movements of the N-terminus, S3 kink and L6 allowing MAH passage from the P-pocket to the mouth of the lateral opening. It seems not unreasonable to propose that the same mechanism could allow forward diffusion from the P-pocket to the lateral opening. Interestingly, the equivalent sphincter residues in CymD are Ala3, Val106 and Ile373, suggesting that the passage from the P-pocket to the lateral opening is narrower in TodX compared to

CymD. We hypothesised that this could contribute to the inability of TodX to transport the larger substrate p-cymene (Fig. 1). To test this, we generated the CymD single variants A3V and V106L, and tested growth on toluene and p-cymene. Both proteins are expressed at wild-type levels (Fig. 8). While the A3V mutant grows identical to wild type, an interesting phenotype is obtained for V106L. Using p-cymene, growth of V106L is indistinguishable from the background ($p = 0.2$), whereas with toluene clear growth above the background is observed ($p < 0.000001$; Fig. 8c). Thus, the V106L mutation in CymD causes a TodX-like phenotype, where growth on p-cymene is abolished and partially restored on toluene, providing support for the importance of the sphincter for substrate diffusion from the P-pocket to the lateral opening. From there, both steered and equilibrium simulations show that MAH can move readily into the OM, even on the short timescales of MD simulations (Fig. 7, Supplementary Fig. 9 and Supplementary Movie). However, while we can identify plausible routes for MAH exit from the lateral gate, the sheer dimensionality of the problem, exacerbated by the LPS in the outer leaflet which moves very slowly, makes it unfeasible to perform an exhaustive energetic study to predict the most likely route(s).

An important question is what happens with the MAH after moving into the OM. The positioning of the lateral opening, at the interface region of the LPS leaflet (Supplementary Fig. 13), allows movement of the hydrophobic molecule into the hydrocarbon layer of the LPS. Calculation of PMF profiles for benzene permeation across the OM has shown that the interface region of the LPS provides a strong energetic barrier[32], preventing movement of the MAH back into the extracellular environment. Inside the hydrophobic core of the OM, the benzene is very mobile and readily enters the inner, phospholipid leaflet (Supplementary Fig. 13). The interface of the phospholipid leaflet forms a much smaller energetic barrier for benzene[32], and we propose that

MAH move into the periplasmic space via mass action, governed by their partitioning coefficients. MAH accumulation into the OM is likely to be tolerated relatively well compared to the IM, for which harmful changes in structure and function as a result of MAH are well-documented[16]. An advantage of the mass-action model is that it may provide a steady, low-level of MAH flux into the IM, preventing toxicity. Alternative, and in our opinion less likely, possibilities for FadL substrates to reach the IM are protein-protein[33] and perhaps lipid-lipid bridges[34] connecting the OM and IM.

FadL orthologs are very widespread in Gram-negative bacteria and, according to the Transporter Classification Database (http://www.tcdb.org/), are found within the phyla Proteobacteria, Nitrospirae, Spirochetes, Chlamydiae and the Bacteroidetes/Chlorobi group. Within the Proteobacteria, FadL orthologs are present in α- through ε-Proteobacteria. Despite their abundance, very few FadL orthologs have been studied, and based on the enormous number of potential substrates it is likely that this family mediates many important processes in Gram-negative bacteria. Our data on MAH uptake provide support for this hypothesis, since they show that small, moderately hydrophobic molecules also function as substrates of FadL channels. Quorum sensing is one potential area in which FadL-mediated transport might play a major role, given that many quorum sensing/autoinducer molecules are hydrophobic[35]. Indeed, FadL orthologs have been shown to mediate N-acyl homoserine lactone uptake in *Rhizobium* spp[36]. Many *Vibrio* spp., commonly studied as quorum sensing model organisms, possess multiple FadL orthologs (*V. cholerae* has three), and it is tempting to speculate that some may be involved in quorum sensing signal uptake. Another, completely unexplored area concerns the potential role of FadL channels in the human gut microbiota. Many species within the abundant genus *Bacteroides* have several FadL orthologs (*B. thetaiotaomicron* has four). In addition, the gut contains a huge arsenal of potential substrates, such as LCFA, bile acids and sterols, suggesting that FadL channel-mediated processes could be important in gut biology.

## Methods

**Strain construction**. Mini-Tn7 elements can be transferred readily into different backgrounds, since the molecular biology of Tn7 is well understood. The TnsABCD transposase components of Tn7 catalyse site- and orientation-specific insertion with high frequency into bacterial chromosomes at Tn7 attachment (attTn7) sites. These sites are located downstream of the highly conserved *glmS* gene, encoding essential glucosamine-6-phosphate synthetase (*Pput*_5291 in *Pp*F1)[37]. Tn7-based genome knock-ins of *Pp*F1-3 were performed according to established procedures[38]. Briefly, full-length wild-type and mutant genes (including their native signal sequence) were cloned in the multiple cloning site (MCS) of the pTJ1 delivery vector, which contains the mini-Tn7T-Tp-*araC*-P_BAD-MCS mobile element for cloning and subsequent integration into the chromosome; genes cloned in this vector are under the control of the P_BAD arabinose-inducible promoter. The plasmid also contains the *dhfRII* gene encoding trimethoprim resistance, which is used to select for chromosomal integration. The pTJ1 vector was moved into the S17 λ-pir conjugation strain via heat shock of competent cells. Plasmid transfer to *Pp*F1-3 was achieved via 4-parental conjugation. Additional *E. coli* strains used were SM10 (λ-pir) containing the helper plasmid pTNS3 encoding the transposase gene, and SM10 (λ-pir) containing the conjugation helper plasmid pRK2013[38]. The helper plasmids were moved into SM10 cells via electroporation at 2500 V (2 mm cuvette). Subsequently, 5 ml overnight cultures of the three plasmid-containing *E. coli* strains were set up in LB with appropriate antibiotics at 37 °C, in addition to the *Pp*F1-3 strain at 30 °C in LB. The next morning, *Pp*F1-3 was sub-cultured at 1:50 dilution and *E. coli* at 1:100 dilution (+ antibiotics) and grown until late log phase (OD ~ 1; ~4–5 h). 250 μl of *Pp*F1-3 culture and 250 μl of each of the *E. coli* cultures were combined, centrifuged at 5,000 rpm for 3 min and resuspended in 100 μl LB. The cells were spotted onto a pre-warmed dry LB plate (no selection) and incubated at 30 °C overnight. The spot was collected in 1 ml LB, centrifuged and resuspended in 100 μl LB. Finally, cells were plated out on Pseudomonas Isolation agar plates with 1.5 mg/ml trimethoprim and incubated overnight at 30 °C. Colony PCR was performed to check for Tn7 integration[37]. The following Tn7-complemented strains were made in *Pp*F1-3: *todX*, *cymD*, F1*fadL*, *todX* ΔS2, *todX* ΔS2S3, *cymD* ΔS2, *cymD* ΔS3, and *cymD* G105A.

For plasmid-based complementation, *cymD* constructs containing a C-terminal His-tag were cloned in the pHERD30 plasmid, a broad-host-range expression vector that utilises the pBAD promoter and the *araC* regulator[39]. The MCS of pHERD vectors is the same as that in the pTJ1 delivery vector described above, facilitating transfer of constructs between both systems. Plasmids were maintained in *E. coli* in the presence of 15 μg/ml gentamycin and in *Pp*F1-3 with 60 μg/ml gentamycin. The following *cymD* constructs were made: wild type, ΔS2, ΔS3, ΔS2S3, G105A, A3V, V106L, L164D/L166D and L273N/F276D.

All mutants were obtained via standard site-directed mutagenesis using the Q5 system (New England Biolabs), according to manufacturer specifications. For TodX ΔS2, the native sequence [67]PKSSTR**SNNRAP**YVGPQ was replaced with PKSSTYAVGAQ. Thus, seven residues (underlined) from the S2 bulge were removed, an alanine was inserted to bridge the expected gap, and a proline at the end of the bulge was replaced by an alanine. For the TodX ΔS2S3 variant, the conserved G104 was deleted within the ΔS2 background. For CymD ΔS2, the sequence [67]KADSHS**RGRNNG**PYVAPE was replaced by KADSHSYAVAAE, analogous to TodX (mature sequence numbering). To generate the ΔS3 strain, G105 was removed (and replaced by Ala in the G105A strain). The L273**N**/F276D mutant was designed to be L273**D**/F276D, but after sequencing the only correct, in-frame mutant that was obtained corresponded to L273N/F276D.

**Plate growth assays**. Plates were made using standard LeMasters-Richards (LR) minimal medium without any carbon source and 1.5% regular agar. For pHERD-based complementation, the plates included 60 μg/ml gentamycin. Arabinose was added in concentrations ranging from 0–0.5% (w/v). For each growth experiment, several colonies were picked from freshly made LB plates containing the respective *Pp*F1-3 strains and resuspended in 0.5 ml LR medium and starved for 15 min at room temperature. 2–3 μl of resuspended cells were spotted on each plate and streaked out. The plates (up to four at a time) were incubated in a closed, plastic Tupperware box that also contained a microcentrifuge tube with 150 μl of either toluene or p-cymene. The tube was closed in the case of toluene, to prevent excessively fast evaporation. Plates were incubated at 30 °C for up to 5 days in the case of p-cymene and at room temperature for up to 4 days in the case of toluene. Colony sizes were measured using a stereo microscope. Values reported are for the largest, isolated colonies observed, typically present at the front of the streak.

**MAH channel expression and purification**. Due to low yields for OM-expressed protein, TodX mutants were expressed as inclusion bodies (IB) in *E. coli*, followed by denaturation and in vitro folding as previously described[25]. For expression of TodX in IB, plasmid ptodX(ib) was constructed by cloning the mature *todX* gene sequence (corresponding to residues 22–453 and with a methionine added at the N-terminus), with a C-terminal hexahistidine tag, downstream of the arabinose-inducible promoter on pBAD22. In this construct, the N-terminal threonine is replaced with a methionine in order to keep to length of the N-terminal plug domain constant. The vectors for the ΔS2 and ΔS2S3 mutants were obtained via standard site-directed mutagenesis using the Q5 system (New England Biolabs), according to manufacturer specifications. For ΔS2, the native sequence [67]PKSST**RSNNRAP**YVGPQ was replaced with PKSSTYAVGAQ. For the ΔS2S3 variant, the conserved G104 was deleted within the ΔS2 background. Protein expression was carried out in Bl21 (DE3) cells at 37 °C in LB/Amp (100 μg/l) by growing the cells to an OD600 of ~0.6–0.8, followed by induction with 0.1% (w/v) arabinose for 2.5–3 h at 37 °C. Following this, cells were harvested by centrifugation, resuspended in TSB buffer [20 mM Tris, 300 mM NaCl, pH 8] and ruptured by one passage at 23,000 psi through a cell disrupter (Constant Systems 0.75 kW). Inclusion bodies and cell debris were collected by centrifugation at 17,000 × *g* for 20 min. The pellet was resuspended by homogenisation in 10 mM Tris (pH 8), 50 mM NaCl, 1% Triton X-100 and stirred for 30 min at room temperature. This wash step was repeated twice with buffer lacking Triton X-100. The washed pellet was resuspended in 10 mM Tris (pH 8), 50 mM NaCl containing 8 M urea, and the suspension was stirred for 2 h or overnight at room temperature. After centrifugation (46,000 × *g*, 30 min), the supernatant containing urea-denatured protein was slowly (dropwise) diluted 10-fold in 1% (wt/vol) *N*-lauroylsarkosine (sarkosyl) in TSB with fast stirring at room temperature. Protein folding was achieved by slowly stirring the sarkosyl solution at 4 °C for ~18 h. Folded TodX protein was purified by standard nickel affinity and gel filtration chromatography as described for FadL[23]. Success of the sarkosyl folding was confirmed by observing the heat-modifiability on SDS/PAGE gels; before boiling, TodX migrates on SDS/PAGE gels with an apparent molecular mass of ~28 kDa, and after boiling, the migration shifts to ~46 kDa.

In contrast to TodX, CymD expressed to reasonable levels within the *E. coli* OM. For expression, the mature part of *cymD* (residues 26–460) was cloned into a modified pET22b vector containing the E. coli FadL signal sequence (residues 1–26). The resulting construct has a C-terminal hexahistidine tag for purification via IMAC. Expression was achieved in C43 (DE3) cells in which the *cyoABCD* operon (Cytochrome bo oxidase) on the chromosome has been deleted, facilitating purification of membrane proteins post-IMAC that are otherwise contaminated with the Cyo complex. Cells were grown in LB/Amp to an OD of ~0.4 at 37 °C, after which the temperature was lowered to 18 °C, and cells were induced 30 min later with 0.2 mM IPTG. Cells were harvested by centrifugation, resuspended in TSB buffer (20 mM Tris, 300 mM NaCl, pH 8) and ruptured by one passage at

23,000 psi through a cell disrupter (Constant Systems 0.75 kW). Total membranes were obtained by ultracentrifugation in a Beckman 45Ti rotor for 50 min at 205,000 × $g$ at 4 °C. Membranes were extracted via homogenisation in 3% Elugent in TSB (using 150 ml for cells from a 12 l culture) for 2 h or overnight, followed by ultracentrifugation (45Ti rotor for 30 min at 205,000 × $g$ at 4 °C). The extract was loaded onto a ~10 ml nickel column (chelating sepharose; GE Healthcare) equilibrated in 0.15% LDAO in TSB. The column was washed with 15 column volumes LDAO buffer containing 25 mM imidazole and subsequently eluted with three column volumes buffer with 200 mM imidazole. After concentration by ultrafiltration to ~3 ml (MWCO 50 kD; Amicon Ultra-15; Millipore) the IMAC step was followed by Superdex-200 16/60 gel filtration in 10 mM Hepes/100 mM NaCl/0.05% LDAO pH 7.5. For crystallisation, another gel filtration step was performed in which the LDAO was replaced by 0.4% $C_8E_4$. Peak fractions were pooled, concentrated to ~15 mg/ml, and aliquots were flash-frozen in liquid nitrogen. The final yield of CymD was ~2–2.5 mg/l cell culture.

To assess expression levels of *cymD* variants, electrocompetent cells of *Pp*F1 in 10% glycerol at 4 °C were electroporated with pH30 plasmids (~100 ng DNA; 2 mm cuvette, 2500 V). Cells were recovered for 1 h at 30 °C in LB and plated out on LB with 60 μg/μl gentamycin and incubated overnight at 30 °C. Several colonies were picked from plate and used to inoculate 10 ml LB with 60 μg/μl gentamycin, which was grown at 30 °C overnight and used to inoculate 0.5 liter cultures in the morning. Strains were grown at 30 °C, induced with 0.5% arabinose at $OD_{600}$ values between 0.4 and 0.8, and grown for another ~4 h. Cells were harvested by centrifugation and lysed via a cell disrupter as described above. Following membrane collection via ultracentrifugation, the pellets were homogenised in 20 ml TSB buffer with 1.5% LDAO and stirred for 45 min at 4 °C. After ultracentrifugation, the clarified membrane extract was loaded onto a 2.5 ml Ni column equilibrated in TSB with 0.15% LDAO. The column was washed with 40 ml buffer containing 30 mM imidazole and eluted in 4–5 ml buffer with 200 mM imidazole. The elution fractions were analysed via SDS-PAGE. The gel pictures reported are representative of two independent expression analyses. The reason for not using western blots to assess expression levels is that, for reasons that are not clear, the detection of both TodX and CymD via anti-His Westerns is very poor and not reproducible.

**Crystallisation and structure determination**. Proteins were crystallised via sitting-drop vapour diffusion, using in-house and commercial screens to obtain initial hits (MemGold1, MemGold2, Morpheus; Molecular Dimensions) via a Mosquito crystallisation robot (TTP Labtech) at 20 °C. Where needed, hits were optimised by fine screening of sitting-drop or hanging-drop setups with larger volumes. Both for TodX mutants and CymD a number of initial hits were obtained, but after optimisation only a limited number diffracted to useful resolutions (~4–5 Å or better on a home source). For TodX ΔS2, the best crystals grew in 0.1 M Na-citrate (pH 3.5), 0.2 M Li$_2$SO$_4$, 28% PEG 400. For TodX ΔS2S3, conditions were similar (0.05 M Na-citrate (pH 4.5), 0.1 M Li$_2$SO$_4$, 30% PEG 400). For CymD, the best crystals were obtained in optimisations from Morpheus F12 (0.1 M bicine/Trizma base pH 8.5, 0.02 M mixed monosaccharides, 10-15% PEG 1000/10-15% PEG 3350/12.5% MPD). Diffraction data were collected at the Diamond Light Source (Didcot, UK) and auto-processed via xia2[40] or DIALS (version 3.1)[41]. The structures were solved by molecular replacement with Phaser[42] within Phenix[43,44], using wild-type TodX (PDB 3BRZ) as the search model (Supplementary Fig. 5). The structures were built via iterative model building cycles within COOT[45] and refinement with Phenix, using TLS refinement for the B-factors. All structures contained 1 molecule in the asymmetric unit, with solvent contents ranging from ~70% for both TodX mutants to ~50% for CymD. As is the case for wild-type TodX, and for unknown reasons, the electron density from all crystals is relatively poor. This is illustrated by high average B-factors of ~128 Å$^2$ for TodX ΔS2 and ΔS2S3, and ~76 Å$^2$ for CymD. Not surprisingly, this led to relatively high numbers of outliers in the Ramachandran plots and relatively high R-factors (Supplementary Table 1). The Ramachandran statistics are (favoured/outliers %): TodX ΔS2, 82.1/4.6 %; TodX ΔS2S3, 87.1/3.9 %; CymD 94.1/1.1%. Structure validation was performed with MolProbity[46].

**System preparation for molecular dynamics**. The crystal structures of TodX (PDB 3BRZ)[25], TodX ΔS2, TodX ΔS2S3 and CymD were completed by adding in the missing residues using Modeller 9.02 (https://salilab.org/modeller/)[47,48]. In order to mimic its natural environment, the completed protein structure was then embedded in an equilibrated *E.coli* model outer membrane containing Ra-LPS molecules in the outer leaflet and a mixture of 1-palmitoyl 2-cis-vaccenic phosphatidylethanolamine, 1-palmitoyl 2-cis-vaccenic phosphatidylglycerol, and 1-palmitoyl 2-cis-vaccenic 3-palmitoyl 4-cis-vaccenic diphosphatidylglycerol (cardiolipin) in the ratio 18:1:1 in the inner leaflet[49–52], developed by Piggot et al.[53], using the g_membed protocol[54]. The Gromos 54a7 force-field was used[55]. The protein-membrane complex was solvated by the simple point charge (SPC) water model[56] and neutralising counterions. To relax any steric conflicts within the system generated during set up, the system was subjected to energy minimisation of 1000 steps using the Steepest Descents method. An equilibration procedure followed in which the entire protein was subjected to positional restraints (with force constant 1000 kJ mol$^{-1}$ nm$^{-1}$) for 10 ns of MD, followed by 10 ns in which only the protein backbone was restrained. Visual molecular dynamics (VMD)[57] was

used as a tool to place benzene molecules around the entrance of TodX transporter for the initial simulations. Initial benzene location in subsequent simulations is explained where the specific methods for those simulations are discussed.

**Equilibrium MD simulations**. GROMACS 5.1.4[58] was employed for all simulations. The short-range electrostatic and vanderWaals cutoffs were set to 1.4 nm, whereas the long-range electrostatic interactions were treated using the particle mesh Ewald (PME) method[59]. All atoms were constrained using the LINCS algorithm[60] to allow the time step of 2 fs. Temperature was kept at 310 K using the V-rescale thermostat[61]. To maintain a constant pressure at 1 atm, the Parrinello-Rahman barostat[62] was applied with a time constant of 1 ps using semi-isotropic coupling. Equilibrium MD systems were run with at least two repeats, each starting with different initial velocities. The results were analysed using functions within the GROMACS package[58]. Molecular graphic images were prepared using both VMD and Pymol[63].

**Steered MD simulations**. Constant velocity pulling simulations were performed along the ±Z direction (principle axis of the protein). The initial benzene position was extracted from the equilibrium MD simulations. A harmonic spring with force constant of 1000 kJ mol$^{-1}$ nm$^{-2}$ was attached to the benzene and pulled at a rate of 0.5 nm ns$^{-1}$ for ~12 ns (six independent simulations were performed to calculate each force curve from steered MD simulations). The force constant and the pulling speed were chosen based on our previous work on arginine permeation through OprD[64]. Snapshots along the ±Z directions were then selected as initial starting points for umbrella sampling windows, such that they were separated by 0.1 nm between windows. Each window was initially equilibrated with a restraint on the benzene for 1 ns. Each equilibrated system was then subjected to 200 ns MD simulation by applying a harmonic biasing potential on the centre of mass of the benzene; a harmonic force constant of 1000 kJ mol$^{-1}$ nm$^{-2}$ was used to restrain the benzene along the Z direction, and no restraints were applied in the *xy* plane. In the case of insufficient sampling, the extra umbrella sampling windows were added based on visual inspection of histogram overlap. The PMF profiles were then computed using Weight Histogram Analysis Method (WHAM)[65] as implemented within GROMACS. The estimated correlation time for each window and the Bayesian bootstrapping error of the PMF was also calculated using the built-in GROMACS tools[66]. The non-cyclised PMF profile is shown in Supplementary Fig. 6. Steered MD simulations in which the benzene was pulled out through the lateral gate were performed such that the benzene was restrained along a vector from the lateral gate of the protein to the outer membrane.

**Reporting summary**. Further information on research design is available in the Nature Research Reporting Summary linked to this article.

## Data availability
The data supporting the findings of this study, including simulation snapshots and parameters are available from the corresponding authors upon reasonable request. Coordinates and structure factors that support the findings of this study have been deposited in the Protein Data Bank with accession codes 6Z34 (CymD), 6Z37 (TodX ΔS2) and 6Z38 (TodX ΔS2S3). The coordinates of TodX and CymD with the missing parts modelled have been deposited in Zenodo. Source data are provided with this paper.

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

## Acknowledgements

This work was supported by a NIH RO1 grant (1R01GM104495-01 to BvdB). BvdB was also supported by a Royal Society Wolfson Merit award. K.S. is funded by the Development and Promotion of Science and Technology Talents Project (DPST), Thailand. The authors wish to thank Mark Sansom for helpful discussions, in particular concerning the free energy calculations. The authors acknowledge the use of the IRIDIS High Performance Computing Facility, and associated support services at the University of Southampton and the use of the UK national supercomputer, ARCHER granted via the UK High-End Computing Consortium for Biomolecular Simulation, HECBioSim, supported by EPSRC (grant no. EP/R029407/1), in the completion of this work. S.K. is supported by the same grant.

## Author contributions

MD simulations were performed by K.S., supervised by S.K. A.D. constructed strains, performed assays together with D.B., purified and crystallised proteins, and determined crystal structures. A.B. collected X-ray crystallography data. B.v.d.B. initiated the project, constructed strains, performed the reported growth assays, and determined crystal structures. The manuscript was written by B.v.d.B. and S.K.

## Competing interests

The authors declare no competing interests.
