## [Peer Review File · Nature Communications]

REVIEWER COMMENTS

Reviewer #1 (Remarks to the Author):

Somboon, et al. describe structural, simulation, and functional studies of two proteins implicated in the uptake of monoaromatic hydrocarbons (MAH) in *Pseudomonas putida*: TodX and CymD. Both of these proteins are outer membrane porin proteins related to FadL, which is involved in the initial uptake of fatty acids, suggesting that TodX and CymD may operate in a similar way. The authors show in cell-based assays that TodX and CymD can both mediate toluene utilization (presumably allowing uptake across the outer membrane). They determined the structures of CymD, and of TodX mutants engineered to restrict a lateral gate (proposed to allow the exit of MAH from the porin lumen out into the outer membrane), and that these mutants indeed impair the ability of cells to utilize MAH as a carbon source. Finally, they use MD simulations to explore possible MAH binding sites, conformational changes in the proteins, and the transport pathway from the extracellular space through TodX/CymD to the periplasm vs the lateral gate.

Overall, I think this is a solid piece of work, shedding light on the uptake of MAH. Understanding this process may aid in the development improved proteins/strains for bioremediation/biodegradation of toxic compounds, and bacterial transport processes more broadly. I think the manuscript is appropriate for the journal, technically sound in the areas I am competent to assess (structures and outer membrane biology) and could be accepted and published essentially as-is. I have no major comments or concerns, though I have included some minor comments that I hope will be helpful to the authors in finalizing their manuscript.

Major Comments:

None.

Minor Comments:

The authors use a mixture of AH and MAH in the intro. If they are not making a distinction between different kinds of compounds, perhaps it would be better to use one or the other throughout?

Use of benzene in MD simulations: The authors state in line 167-168 that benzene is a substrate for the bacterial strain PpF1, they don't show data that benzene is a substrate of TodX, and don't cite a reference. While I understand the logic to some degree (benzene is similar to toluene), they also show that CymD can discriminate between toluene and p-cymene, which are also similar. Since they later do essentially the same simulations using p-cymene, why not present those results front and center, and if desired, include the benzene analysis later to support it? Along these lines, my impression from Fig. S2 is that the benzene binding site is not well defined by simulation, yet the selectivity of FadL family members would suggest that ligand binding would be better defined. Would it be worth trying to estimate the binding energy of the observed poses to try to assess if any of them stand out from the pack?

Is there any experimental evidence that "state II" is populated to a significant degree (with or without ligand bound)?

Validation of MD simulations: Can any of the hypotheses raised by the MD simulations be tested experimentally? For example, if the P-pocket is indeed a substrate binding site, mutations here should attenuate growth in the complementation experiments. Similarly, several residues are proposed to stabilize "state II" of the N-terminus. If this conformation indeed exists and is important for function, mutations in these residues might be expected to impact function.

Structures: Some stats from the Table S1 and the validation reports initially raised some concerns

about the model refinement (e.g., large number of Ramachandran outliers and rather large gaps between R_{work} and R_{free} for all three structures ($\sim 6\%$)). However, as indicated, in the methods, this has been an issue with other TodX structures (and a quick check of the PDB reflects this). The last author is an experienced crystallographer, and I am inclined to believe they have done the best they can with the datasets that they have.

Ln 301-302: I didn't follow the logic here. Was there a specific concern about CymD expression levels, but not TodX?

Fig. 5e: Since the proteins are all tagged anyway, perhaps an anti-His Western would be more clear? This is completely optional, but would just be cleaner/more specific, and could maybe even just be done in total lysates/membrane fractions, which would introduce less perturbation on the apparent expression levels than going through a complete purification.

Ln 407-417: I am having a little trouble figuring out what exactly is being done here. Are the authors simulating p-cymene in TodX or CymD, or a mix of the two?

In 419-440: How does lateral exit of p-cymene from CymD compare to benzene exiting TodX? It is not clear to me if p-cymene uses path A, B, C, or something different.

Reviewer #2 (Remarks to the Author):

Somboom et al. study the uptake mechanism of monoaromatic compounds (MAH) by outer membrane proteins FadL channels. Based on growth assays, molecular dynamics and crystallography, they conclude that MAHs do not pass the channel along the "classical" route (straight along the beta barrel) but across a lateral opening towards the hydrophobic membrane interior. This "lateral diffusion" route has been found before for fatty acid permeation across E coli FadL by one of the authors (Van den Berg and coworkers, Nature 2009). The present study shows convincingly that this uncommon pathway is also used by monoaromatic hydrocarbons.

This study is overall a well conducted and insightful combination of simulations, structural biology, and growth assays. What makes this study exceptional is coverage of scales from living organisms down to atomic-level mechanisms and free energies. Such studies are still rare. The lateral opening in other FadL channels, however, has been characterized before, albeit for other apolar compounds (fatty acids instead of aromatics).

Before publication, some additional information and analysis is needed:

1) It was not clear to me whether the reduced growth for the F1-3 strain ("background growth", relative to F1, Fig 1c) is caused purely due to the lack of carbon uptake or whether the triple knockout can have other side effects that could impede the growth. If you could restore the growth with another carbon source that does not depend on F1, or restore the wt growth with increased toluene concentration, such side effects could be excluded.

2) The equilibrium simulations did not sample the transition from the P-pocket to the lateral opening indicative of a substantial energy barrier. Therefore, to justify the conclusion that the lateral opening is main exit pathway, additional simulations are needed. So far, the authors write

"The limitation of equilibrium MD timescales is not allowing us to observe this movement, and likely requires enhanced sampling methods in follow-up studies."

and

"It seems not unreasonable to propose that the same mechanism could allow forward diffusion from the P-pocket to the lateral opening."

Enhanced sampling can be used (and was used by the authors for the classical route), and the second statement can be tested. I suggest computing the free energy profile (PMF) for the P-pocket-lateral gate transition, e.g. with umbrella sampling. The barrier should then be compared with the barrier for the classical pathway. In case that umbrella sampling does not work well, the authors could also try metadynamics.

Note that the barrier for the classical pathway is quite low (only $\sim 7kT$, $\sim 17kJ/mol$), so from the PMF alone this route can not be excluded. Hence the statement "indicate that the classical pathway is not a feasible" (line 331) should be revised.

Minor:

1) Some labels and/or legends in Figure 1c would help the reader a lot (color code for bars, labels with 20 and 30 degree C, and same for Fig. 5b/d). Likewise, the respective paraph (lines 143-156) would benefit from more precise references to subpanels in Fig 1 (e.g. "Fig 1c" -> "Fig 1c, white bars" or similar).

2) During the umbrella sampling simulations for the PMF in Fig 6, was some restraint applied to the solute in bulk solvent, so it would not "miss" the pore?

3) For the PMF in Fig. 6, the free energy in the bulk (at $+45A$) was defined to zero. This procedure is often used, but can also hide severe convergence problems (see e.g. Golla et al., JCTC 16, 2751-2765 (2020)). Please report original PMF also in the SI.

4) I did not find how the statistical errors in the force profiles were computed. Are they just standard deviations, or standard errors? How were they computed? Errors on the PMFs are missing.

Reviewer #3 (Remarks to the Author):

The manuscript by Somboon et al. is well written and organized. They present a series of experimental and computational approaches supporting that monoaromatic hydrocarbons (MAH) are taken up by lateral diffusion in *Pseudomonas putida* F1 (PpF1) by means of the FadL-like channels TodX and CymD.

The authors start by creating a triple mutant of PpF1 lacking its 3 FadL orthologous, FadL, TodX and CymD. The growth of such mutant on toluene or p-cymene vapours is severely reduced with respect to the growth shown by the wild type strain. CymD can complement growth on toluene and p-cymene, while TodX can only complement for growth on toluene. FadL cannot complement growth in any of the two compounds. These results corroborate previous data indicating that FadL channels are substrate-specific. Using equilibrium molecular dynamics (MD) simulations with TodX and benzene, they identified that an specific conformation of TodX is required to localize benzene in the P-pocket near the N-terminus. Following a similar strategy as that previously described for EcFadL, lateral opening mutants of TodX and CymD were created. Variants of TodX were created with a flattened S2 beta-strand or with S2 and S3 flattened strands. The crystal structures of the two mutant channels $\Delta S2$ and $\Delta S2S3$ showed reduced lateral opening while retaining the general structure previously described for the wild type TodX protein. In this study, a preliminary structure of CymD was obtained that shows a similar structure to TodX and variants of CymD lacking S2 or S3 were created. Experimental evidence was obtained showing that lateral openings are required for uptake of MAH

since the triple mutant PpF1 could not be complemented for growth in MAH when the strain carries lateral opening mutants of TodX or CymD. They used chromosomal and plasmid-based complementation. Finally they provide additional support for the lateral diffusion mechanism uptake of MAH via TodX and CymD with a series of umbrella sampling and steered MD simulations.

I recommend this paper for publication in Nature Communications and I have only minor comments:

Line 53, Change "AH" to "Aromatic hydrocarbons (AH)".

Supplementary Fig 1, Include in part "a" structure and coefficient partition for benzene. Benzene is a physiological substrate for PpF1 and it was used in the MD simulations with TodX.

I suggest to include a clear indication of the position of the outer membrane (OM) in the ribbon figures. For example, in Fig. 6 and Fig. 7 there is a slight shadowing showing position of the outer membrane. However, when the PDF is printed, the shadowing is barely visible. Could the authors add thin black lines to better show the extend of the OM?

In Supplementary Fig. 11 the legend says: "...where a pocket is formed by Leu28 and Lys43." However, in the inset of panel (b) is shown Leu48 (not 28). For the case of the Lys residue, in the figure one could read Lys431 instead of Lys43. Which are the correct residues?

Line 558, change "1.5mg/ml" to "1.5 mg/ml". I.e, space after 1.5.

REVIEWER COMMENTS

Reviewer #1 (Remarks to the Author):

Somboon, et al. describe structural, simulation, and functional studies of two proteins implicated in the uptake of monoaromatic hydrocarbons (MAH) in *Pseudomonas putida*: TodX and CymD. Both of these proteins are outer membrane porin proteins related to FadL, which is involved in the initial uptake of fatty acids, suggesting that TodX and CymD may operate in a similar way. The authors show in cell-based assays that TodX and CymD can both mediate toluene utilization (presumably allowing uptake across the outer membrane). They determined the structures of CymD, and of TodX mutants engineered to restrict a lateral gate (proposed to allow the exit of MAH from the porin lumen out into the outer membrane), and that these mutants indeed impair the ability of cells to utilize MAH as a carbon source. Finally, they use MD simulations to explore possible MAH binding sites, conformational changes in the proteins, and the transport pathway from the extracellular space through TodX/CymD to the periplasm vs the lateral gate.

Overall, I think this is a solid piece of work, shedding light on the uptake of MAH. Understanding this process may aid in the development improved proteins/strains for bioremediation/biodegradation of toxic compounds, and bacterial transport processes more broadly. I think the manuscript is appropriate for the journal, technically sound in the areas I am competent to assess (structures and outer membrane biology) and could be accepted and published essentially as-is. I have no major comments or concerns, though I have included some minor comments that I hope will be helpful to the authors in finalizing their manuscript.

Major Comments:

None.

Minor Comments:

The authors use a mixture of AH and MAH in the intro. If they are not making a distinction between different kinds of compounds, perhaps it would be better to use one or the other throughout?

We now only use the term MAH.

Use of benzene in MD simulations: The authors state in line 167-168 that benzene is a substrate for the bacterial strain PpF1, they don't show data that benzene is a substrate of TodX, and don't cite a reference.

We did cite ref. 9 which reports growth of PpF1 on benzene (line 62). We have re-cited this paper in line 169. An important reason for not using benzene as a transport substrate is its volatility, which is even higher than that of toluene, combined with its higher toxicity and carcinogenicity.

While I understand the logic to some degree (benzene is similar to toluene), they also show that CymD can discriminate between toluene and p-cymene, which are also similar. Since

they later do essentially the same simulations using p-cymene, why not present those results front and center, and if desired, include the benzene analysis later to support it?

It is TodX that can discriminate between toluene and p-cymene (CymD transports both; Fig. 1). Since we have more simulations of TodX and benzene and a more thorough computational characterisation of this system, would prefer to retain the order and focus as it is currently.

Along these lines, my impression from Fig. S2 is that the benzene binding site is not well defined by simulation, yet the selectivity of FadL family members would suggest that ligand binding would be better defined. Would it be worth trying to estimate the binding energy of the observed poses to try to assess if any of them stand out from the pack?

It is difficult to get reliable estimates of the binding energy from these simulations. In addition, it may well be that the selectivity arises from differences in the lateral opening or other protein regions (see later comments), and not from a different affinity for the binding pocket. To support this notion we have now pulled p-cymene through the TodX lateral opening, which requires more force for p-cymene than for benzene. This is suggestive of selectivity, in the sense that p-cymene would be less able to move through the lateral opening of TodX. Testing this hypothesis requires free energy calculations (which are very difficult) for a comprehensive characterisation of the energetic differences between the two substrates. However, we have added the new steered MD data to the supplementary information (SI Fig. 10) and have discussed it in the Results section of the manuscript.

Is there any experimental evidence that “state II” is populated to a significant degree (with or without ligand bound)?

There isn't. The electron density of the first ~8 TodX residues tends to be poor or absent and also shows evidence of conformational heterogeneity.

Validation of MD simulations: Can any of the hypotheses raised by the MD simulations be tested experimentally? For example, if the P-pocket is indeed a substrate binding site, mutations here should attenuate growth in the complementation experiments.

This is a useful suggestion, but the main problems here are that (i) we would likely need multiple changes in order to see a growth defect due to the considerable number of hydrophobic residues that make up the P-pocket, and (ii) it is far from clear what replacements to choose. For example, one could envision that by making the pocket more hydrophilic, diffusion through the protein could actually be faster since the substrate could spend less time in the P-pocket. It is a complex problem.

However, to accommodate the reviewer we have made two P-pocket mutants in CymD, each with 2 residues changed simultaneously: V164D/L166D and L273N/F276D (the latter was intended to include L273D, but was found to be L273N after sequencing).

We have tested growth of these CymD mutants on both toluene and p-cymene and the results are shown in the new Figure 8 and discussed in lines 420-433. With toluene, both mutants show growths clearly above background, but are present at much lower levels in the OM compared to WT, indicating that their biogenesis is compromised. These data suggest (i) that

factors other than MAH uptake are growth-limiting, and (ii) that both P-pocket mutants are at least partly functional.

In addition to the P-pocket mutations, we also made two single CymD mutants for the "sphincter" constriction that connects the P-pocket to the lateral opening (SI Fig. 11 and line 492). The idea behind this was to generate "TodX-like" changes in CymD that would make this constriction narrower. This might then impede growth on p-cymene but not on toluene, *i.e.* a phenotype similar to that of TodX might be observed (Fig. 1). While one of these mutants (A3V) is indistinguishable from wild type, the V106L mutant does not grow on p-cymene but does support growth on toluene (Fig. 8). Thus, via a single mutation we have changed the specificity of CymD to resemble that of TodX. This is now described in lines 498-509.

Similarly, several residues are proposed to stabilize "state II" of the N-terminus. If this conformation indeed exists and is important for function, mutations in these residues might be expected to impact function.

Again it is unclear what type of changes to make. In addition, we do not observe state II in CymD, so this state may not be important for MAH channels in general.

Structures: Some stats from the Table S1 and the validation reports initially raised some concerns about the model refinement (e.g., large number of Ramachandran outliers and rather large gaps between Rwork and Rfree for all three structures (~6%)). However, as indicated, in the methods, this has been an issue with other TodX structures (and a quick check of the PDB reflects this). The last author is an experienced crystallographer, and I am inclined to believe they have done the best they can with the datasets that they have.

We thank the reviewer for his/her understanding.

Ln 301-302: I didn't follow the logic here. Was there a specific concern about CymD expression levels, but not TodX?

No, this is a general issue. We performed the expression studies on CymD because its expression in PpF1 is higher than that of TodX, making the bands much easier to see on SDS-PAGE (sentence added, lines 302-303).

Fig. 5e: Since the proteins are all tagged anyway, perhaps an anti-His Western would be more clear? This is completely optional, but would just be cleaner/more specific, and could maybe even just be done in total lysates/membrane fractions, which would introduce less perturbation on the apparent expression levels than going through a complete purification.

We agree this might be logical, but for reasons we don't understand the detection of both TodX and CymD via anti-His Westerns is very poor. However, there is no reason to assume that expression levels are differentially affected when samples are generated in identical ways. The expression analyses were also done twice, independently. We have added a section in the methods describing the procedure and rationale (lines 692-707).

Ln 407-417: I am having a little trouble figuring out what exactly is being done here. Are the authors simulating p-cymene in TodX or CymD, or a mix of the two?

We have added “*To explore this, we simulated CymD with p-cymene, in a similar manner to the equilibrium simulations of TodX with benzene described above*” to the text to clarify that we are now discussing simulations of CymD with p-cymene (lines 409-410).

In 419-440: How does lateral exit of p-cymene from CymD compare to benzene exiting TodX? It is not clear to me if p-cymene uses path A, B, C, or something different.

We have modified the legend of SI Figure 9 in which the profiles are shown and also added a main text sentence, to describe the pathway as being somewhat similar to pathway A of benzene exiting TodX (lines 464-466).

Reviewer #2 (Remarks to the Author):

Somboom et al. study the uptake mechanism of monoaromatic compounds (MAH) by outer membrane proteins FadL channels. Based on growth assays, molecular dynamics and crystallography, they conclude that MAHs do not pass the channel along the "classical" route (straight along the beta barrel) but across a lateral opening towards the hydrophobic membrane interior. This "lateral diffusion" route has been found before for fatty acid permeation across E coli FadL by one of the authors (Van den Berg and coworkers, Nature 2009). The present study shows convincingly that this uncommon pathway is also used by monoaromatic hydrocarbons.

This study is overall a well conducted and insightful combination of simulations, structural biology, and growth assays. What makes this study exceptional is coverage of scales from living organisms down to atomic-level mechanisms and free energies. Such studies are still rare. The lateral opening in other FadL channels, however, has been characterized before, albeit for other apolar compounds (fatty acids instead of aromatics).

Before publication, some additional information and analysis is needed:

1) It was not clear to me whether the reduced growth for the F1-3 strain ("background growth", relative to F1, Fig 1c) is caused purely due to the lack of carbon uptake or whether the triple knockout can have other side effects that could impede the growth. If you could restore the growth with another carbon source that does not depend on F1, or restore the wt growth with increased toluene concentration, such side effects could be excluded.

We have added a panel to Fig. 1 showing that the growth of F1, F1-3 and F1-5 on glucose is identical (Fig. 1a).

2) The equilibrium simulations did not sample the transition from the P-pocket to the lateral opening indicative of a substantial energy barrier. Therefore, to justify the conclusion that the lateral opening is main exit pathway, additional simulations are needed. So far, the authors write

"The limitation of equilibrium MD timescales is not allowing us to observe this movement, and likely requires enhanced sampling methods in follow-up studies."

and

"It seems not unreasonable to propose that the same mechanism could allow forward

diffusion from the P-pocket to the lateral opening."

Enhanced sampling can be used (and was used by the authors for the classical route), and the second statement can be tested. I suggest computing the free energy profile (PMF) for the P-pocket-lateral gate transition, e.g. with umbrella sampling. The barrier should then be compared with the barrier for the classical pathway. In case that umbrella sampling does not work well, the authors could also try metadynamics.

It is extremely difficult to define a reaction coordinate to use for umbrella sampling to sample the pathway from the p-pocket to the lateral gate, as geometrically it would involve two steps (down from the p-pocket into the 'middle' of the protein and then across to the lateral gate). Furthermore, as we don't have the same environment on both sides, it would be very difficult to assess convergence for each of the two steps – for the classical route we had bulk water on both ends of the pathway. Doing metadynamics would be a whole new project in itself. The collective variables are not obvious and would require work that would make another paper (and likely PhD).

Note that the barrier for the classical pathway is quite low (only $\sim 7kT$, $\sim 17kJ/mol$), so from the PMF alone this route can not be excluded. Hence the statement "indicate that the classical pathway is not a feasible" (line 331) should be revised.

This is a fair point. We have revised the statement (line 330) to "*indicate that the classical pathway is less favourable for benzene translocation into the periplasmic space*".

Minor:

1) Some labels and/or legends in Figure 1c would help the reader a lot (color code for bars, labels with 20 and 30 degree C, and same for Fig. 5b/d). Likewise, the respective paraph (lines 143-156) would benefit from more precise references to subpanels in Fig 1 (e.g. "Fig 1c" -> "Fig 1c, white bars" or similar).

We have made changes in Figures 1 and 5 to make these issues clearer. Notably, we now use different colours for different strains, with the legend next to the data. We also have shown the strain names rather than numbers on the plates.

2) During the umbrella sampling simulations for the PMF in Fig 6, was some restraint applied to the solute in bulk solvent, so it would not "miss" the pore?

We did not apply any additional restraints, this detail has now been added to the methods (line 776).

3) For the PMF in Fig. 6, the free energy in the bulk (at $\pm 45A$) was defined to zero. This procedure is often used, but can also hide severe convergence problems (see e.g. Golla et al., JCTC 16, 2751–2765 (2020)). Please report original PMF also in the SI.

We have added the non-cyclised PMF to the SI (Fig. 13). The location of the barriers would not change upon additional simulations (and this is corroborated by the close agreement between the PMF and the force profile from steered MD), but we concede there is a possibility that the heights of the barriers may do.

4) I did not find how the statistical errors in the force profiles were computed. Are they just standard deviations, or standard errors? How were they computed? Errors on the PMFs are missing.

Standard deviations are included in on the PMF curves; they are extremely small and have now been made a bit more obvious by darkening the colour of the lines.

The errors on the force profiles are standard deviations, and we have now added this information to the figure legends.

Reviewer #3 (Remarks to the Author):

The manuscript by Somboon et al. is well written and organized. They present a series of experimental and computational approaches supporting that monoaromatic hydrocarbons (MAH) are taken up by lateral diffusion in *Pseudomonas putida* F1 (PpF1) by means of the FadL-like channels TodX and CymD.

The authors start by creating a triple mutant of PpF1 lacking its 3 FadL orthologous, FadL, TodX and CymD. The growth of such mutant on toluene or p-cymene vapours is severely reduced with respect to the growth shown by the wild type strain. CymD can complement growth on toluene and p-cymene, while TodX can only complement for growth on toluene. FadL cannot complement growth in any of the two compounds. These results corroborate previous data indicating that FadL channels are substrate-specific. Using equilibrium molecular dynamics (MD) simulations with TodX and benzene, they identified that a specific conformation of TodX is required to localize benzene in the P-pocket near the N-terminus. Following a similar strategy as that previously described for EcFadL, lateral opening mutants of TodX and CymD were created. Variants of TodX were created with a flattened S2 beta-strand or with S2 and S3 flattened strands. The crystal structures of the two mutant channels deltaS2 and deltaS2S3 showed reduced lateral opening while retaining the general structure previously described for the wild type TodX protein. In this study, a preliminary structure of CymD was obtained that shows a similar structure to TodX and variants of CymD lacking S2 or S3 were created. Experimental evidence was obtained showing that lateral openings are required for uptake of MAH since the triple mutant PpF1 could not be complemented for growth in MAH when the strain carries lateral opening mutants of TodX or CymD. They used chromosomal and plasmid-based complementation. Finally they provide additional support for the lateral diffusion mechanism uptake of MAH via TodX and CymD with a series of umbrella sampling and steered MD simulations.

I recommend this paper for publication in Nature Communications and I have only minor comments:

Line 53, Change “AH” to “Aromatic hydrocarbons (AH)”.

In accordance with reviewer 1’s wishes we have changed ‘AH’ to ‘MAH’ throughout the manuscript.

Supplementary Fig 1, Include in part "a" structure and coefficient partition for benzene. Benzene is a physiological substrate for PpF1 and it was used in the MD simulations with TodX.

Benzene and its partition coefficient have been added to SI Figure 1.

I suggest to include a clear indication of the position of the outer membrane (OM) in the ribbon figures. For example, in Fig. 6 and Fig. 7 there is a slight shadowing showing position of the outer membrane. However, when the PDF is printed, the shadowing is barely visible. Could the authors add thin black lines to better show the extend of the OM?

We have made the extent of the OM clearer in the figures by adding a dashed line.

In Supplementary Fig. 11 the legend says: "...where a pocket is formed by Leu28 and Lys43." However, in the inset of panel (b) is shown Leu48 (not 28). For the case of the Lys residue, in the figure one could read Lys431 instead of Lys43. Which are the correct residues?

We have changed the errant residue number in the caption of SI Fig. 12 (it should be Leu48) and have made the label of Lys431 clearer in the figure.

Line 558, change "1.5mg/ml" to "1.5 mg/ml". I.e, space after 1.5.

This has now been changed.

REVIEWER COMMENTS

Reviewer #1 (Remarks to the Author):

The authors have adequately addressed all my comments, it is a very nice paper.

Reviewer #2 (Remarks to the Author):

The authors have improved the manuscript considerably. Regarding the PMF along the side exit, which I requested in my previous report, I can accept that it is not computed in this work since the simulations mainly corroborate the experimental data (which showed convincingly that the side exit is the predominant route).

But I am seriously concerned about the PMF along the "classical route". In response to my request, now the non-cyclic PMF is shown as SI Figure 13. The PMF is far from converged, as evident from the free energy offset of 22 kJ/mol between the bulk water reservoirs (which would be zero for a converged PMF). I understand that converging such PMFs can be a considerable challenge. However, if the free energy offset is in the same order of magnitude as the reported barrier (17 kJ/mol), we cannot draw any conclusions from the PMF. From the non-cyclic PMF, the barrier could take any value between zero and 20 kJ/mol.

From the non-cyclic PMF it is also clear that the errors now added to the PMF greatly underestimate the true error.

It is unlikely that longer umbrella sampling simulations would solve this issue. Most likely, orthogonal degrees of freedom (such as side chains partly blocking the pore) lead to such problems, as discussed in the work by Golla et al. mentioned in my previous report (JCTC 16, 2751–2765 (2020)). Maybe metadynamics works better here, since metadynamics simulations don't get stuck in a local minimum as often as umbrella sampling. But running metadynamics may indeed be another project.

Since the manuscript greatly benefits from the molecular view given by the simulations, and since the manuscript should get published, the only clean solution would be to remove the PMF. With the evidently problematic PMF, the overall quality of the manuscript would be comprised. The authors could focus on the force profiles instead and switch to a slightly more qualitative discussion.

Reviewer #1 (Remarks to the Author):

The authors have adequately addressed all my comments, it is a very nice paper.

Thank you.

Reviewer #2 (Remarks to the Author):

The authors have improved the manuscript considerably. Regarding the PMF along the side exit, which I requested in my previous report, I can accept that it is not computed in this work since the simulations mainly corroborate the experimental data (which showed convincingly that the side exit is the predominant route).

But I am seriously concerned about the PMF along the "classical route". In response to my request, now the non-cyclic PMF is shown as SI Figure 13. The PMF is far from converged, as evident from the free energy offset of 22 kJ/mol between the bulk water reservoirs (which would be zero for a converged PMF). I understand that converging such PMFs can be a considerable challenge. However, if the free energy offset is in the same order of magnitude as the reported barrier (17 kJ/mol), we cannot draw any conclusions from the PMF. From the non-cyclic PMF, the barrier could take any value between zero and 20 kJ/mol.

From the non-cyclic PMF it is also clear that the errors now added to the PMF greatly underestimate the true error.

It is unlikely that longer umbrella sampling simulations would solve this issue. Most likely, orthogonal degrees of freedom (such as side chains partly blocking the pore) lead to such problems, as discussed in the work by Golla et al. mentioned in my previous report (JCTC 16, 2751–2765 (2020)). Maybe metadynamics works better here, since metadynamics simulations don't get stuck in a local minimum as often as umbrella sampling. But running metadynamics may indeed be another project.

Since the manuscript greatly benefits from the molecular view given by the simulations, and since the manuscript should get published, the only clean solution would be to remove the PMF. With the evidently problematic PMF, the overall quality of the manuscript would be comprised. The authors could focus on the force profiles instead and switch to a slightly more qualitative discussion.

We thank the reviewer for his comments. We have removed the cyclised PMF from Figure 6. However, given that calculation of the PMF is a logical thing to do in this case, we feel that it is still useful to state in the paper that we calculated the PMF, but that due to convergence problems we have not used it in our conclusions. We also feel it is useful for the field to show the non-cyclised PMF, and therefore we have kept Supplementary Fig. 13. In addition, we used the umbrella samplings for placement of the MAH close to the lateral opening (lines 357-364). Since we now mention the non-cyclised PMF in the main text, Supplementary Fig. 13 has become Supplementary Fig. 6 in the revision, and all the SI figures occurring later have been renumbered. We hope that this solution is acceptable. As suggested, the text has also been changed on pages 11 and 12.